# Forms of Community Engagement in Neighborhood Food Retail: Healthy Community Stores Case Study Project

**DOI:** 10.3390/ijerph19126986

**Published:** 2022-06-07

**Authors:** Ravneet Kaur, Megan R. Winkler, Sara John, Julia DeAngelo, Rachael D. Dombrowski, Ashley Hickson, Samantha M. Sundermeir, Christina M. Kasprzak, Bree Bode, Alex B. Hill, Emma C. Lewis, Uriyoan Colon-Ramos, Jake Munch, Lillian L. Witting, Angela Odoms-Young, Joel Gittelsohn, Lucia A. Leone

**Affiliations:** 1Department of Family and Community Medicine, University of Illinois College of Medicine, Rockford, IL 61107, USA; 2Department of Behavioral, Social and Health Education Sciences, Rollins School of Public Health, Emory University, 1518 Clifton Rd, Atlanta, GA 30322, USA; megan.winkler@emory.edu; 3Center for Science in the Public Interest, 1250 I St NW, Floor 5, Washington, DC 20005, USA; sjohn@cspinet.org (S.J.); ahickson@cspinet.org (A.H.); 4Departments of Health Policy Management & Nutrition, Harvard T.H. Chan School of Public Health, Harvard University, Boston, MA 02115, USA; jdeangelo@hsph.harvard.edu; 5Division of Kinesiology, Health and Sport Studies, College of Education, Wayne State University, Detroit, MI 48202, USA; rjankows@gmail.com (R.D.D.); bodebr@wayne.edu (B.B.); 6Department of International Health, Johns Hopkins University Bloomberg School of Public Health, Baltimore, MD 21205, USA; srex2@jh.edu (S.M.S.); elewis40@jhu.edu (E.C.L.); jgittel1@jhu.edu (J.G.); 7Department of Community Health and Health Behavior, University at Buffalo, Buffalo, NY 14260, USA; cmk27@buffalo.edu (C.M.K.); lucialeo@buffalo.edu (L.A.L.); 8Urban Studies and Planning and Detroit Food Map Initiative, Wayne State University, Detroit, MI 48202, USA; alexbhill@wayne.edu; 9Milken Institute School of Public Health, George Washington University, 950 New Hampshire Avenue, Washington, DC 20052, USA; uriyoan@gwu.edu (U.C.-R.); llqw@gwu.edu (L.L.W.); 10School of Natural Sciences and Mathematics, The University of Texas at Dallas, Richardson, TX 75080, USA; jake.munch@utdallas.edu; 11Division of Nutritional Science, College of Human Ecology, Cornell University, Ithaca, NY 14853, USA; odoms-young@cornell.edu

**Keywords:** community engagement, case study approach, cross-case analysis, healthy food retail, retail food environment, community food stores

## Abstract

Community engagement is well established as a key to improving public health. Prior food environment research has largely studied community engagement as an intervention component, leaving much unknown about how food retailers may already engage in this work. The purpose of this study was to explore the community engagement activities employed by neighborhood food retailers located in lower-income communities with explicit health missions to understand the ways stores involve and work with their communities. A multiple case study methodology was utilized among seven retailers in urban U.S. settings, which collected multiple sources of data at each retailer, including in-depth interviews, store manager sales reports, store observations using the Nutrition Environment Measures Survey for Stores, public documents, and websites. Across-case analysis was performed following Stake’s multiple case study approach. Results indicated that retailers employed a wide variety of forms of community engagement within their communities, including Outreach, Building Relationships through Customer Relations, Giving Back, Partnering with Community Coalitions, and Promoting Community Representation and Inclusiveness. Strategies that built relationships through customer relations were most common across stores; whereas few stores demonstrated community inclusiveness where members participated in store decision making. Findings provide a more comprehensive view of the ways local food retailers aim to develop and sustain authentic community relationships. Additional research is needed to evaluate the impact of community engagement activities on improving community health.

## 1. Introduction

The risk of chronic diseases is higher among people living in low-food-access neighborhoods [1,2]. However, improving food infrastructure to facilitate access may not always alter food consumption [3,4]. Engaging communities is increasingly being put forth to improve public health [5] and may be necessary to identify how to make local food environments more health promoting in a way that best serves a community.

Community engagement is “the process of working collaboratively with and through groups of people affiliated by geographic proximity, special interest, or similar situations to address issues affecting the wellbeing of those people” [6] (p. 9). It allows for a better understanding of the needs, culture, and behavior of a specific population [5] and is vital for successful public health interventions [7]. The understanding of community engagement in research has evolved over time, with relationship building becoming a primary focus to ensure maximized participation and sustainable change [5,8]. Community engagement has been conceptualized by the Centers for Disease Control and Prevention (CDC) as occurring across a continuum of five phases: (1) Inform/Outreach, (2) Consult, (3) Involve, (4) Collaborate and (5) Shared leadership [5]. As community engagement progresses across the spectrum, community involvement in shared decision making, collaborative leadership, trust, and communication increases [5]. The key to reaching shared leadership is authentically engaging with the community and remaining grounded in the principles of “co-creation of solutions” and “trust-based relationships” [9].

Extensive literature has demonstrated the effectiveness of public health interventions that collaborate and empower communities to address health disparities [7,10,11,12,13]. The importance of community engagement has been particularly highlighted in under-resourced and/or disadvantaged groups [10,14,15], by providing opportunities of two-way knowledge exchange to further address health disparities [16]. Previous dietary interventions that address the retail food environment [17,18,19,20] have described utilizing different activities to engage communities, ranging from informing (e.g., use of social and local media [18,20]) to community involvement (e.g., cookbook development [17], recipe sharing [18] and cooking demonstrations [19,20]). While these studies have shed light on community engagement as a tool in retail food interventions, there remains a lack of understanding of what community engagement looks like beyond the context of an intervention. The strategies and activities retailers already employ to engage with members of their local community and how this varies across community food stores remains largely unknown. Further, as community engagement facilitates a better understanding of community needs and preferences [16], exploring which factors facilitate and/or impede it is key to knowing leverage points that can enhance retailer-community relationships.

With the goal of informing future retail-based public health interventions, this study explored what community engagement looks like among neighborhood food retailers who aim to provide healthy foods to low-income communities. Specifically, we examined the types of activities planned and employed by retailers as well as the circumstances and processes that facilitated more collaborative forms of engagement. By making these comparisons, we aimed to understand what steps food retailers should consider developing and maintain authentic relationships with their community that will best serve their needs.

## 2. Materials and Methods

### 2.1. Study Design

This study derives from a larger case study project [21] conducted in partnership with members of the Healthy Food Retail working group of the Robert Wood Johnson Foundation’s Healthy Eating Research (HER) program and the CDC Nutrition and Obesity Policy Research and Evaluation Network (NOPREN). The larger case study aimed to examine the experiences of community food stores in providing healthy food in low-income neighborhoods as well as challenges the stores faced during the COVID-19 pandemic. For the project, a multiple case study methodology was employed because it provided in-depth information on selected cases utilizing a mixed-methods approach (i.e., both qualitative and quantitative data collection methods) and permitted the use of multiple data sources to highlight the distinct context within each case. Additional details regarding the study protocol are available elsewhere [21].

### 2.2. Recruitment and Data Collection

The inclusion criteria for each community food store to participate in the larger study included: serving low- or low-to-middle-income communities, open for at least one-year, clear mission statement to improve healthy food access, willing to share stocking, sales and purchasing data, and acceptance of Women, Infants, and Children (WIC) and/or Supplemental Nutrition Assistance Program (SNAP) benefits. A total of seven community food stores in different cities across the U.S. were selected by the NOPREN Healthy Food Retail workgroup using a maximum variation sampling approach [22]. Additional information regarding selection of stores has been previously published [21]. To maintain anonymity of each store (also known as *case* in this study), we refer to each case by the city in which the store is located. Retailers in this study were located in: Baltimore, MD; Boston, MA; Buffalo, NY; Chicago, IL; Detroit, MI; Minneapolis, MN; and Washington, DC.

Several data sources were utilized, including in-depth interviews, store manager sales reports, Nutrition Environment Measures Survey for Stores (NEMS-S), public documents, and websites. In-depth interviews were one of the primary methods that provided information on community engagement and/or partnerships among stores within their surrounding communities. Two different interview guides were developed each for store leadership and for store stakeholders, which included community members and organizations as well as store vendors and distributors. Each site interviewed on average 3–4 store leaders and 3–4 stakeholders. To collect information around community engagement, the stakeholder’s guide had questions around the history of the partnership between stores and stakeholders, and the retailer’s guide asked leadership how they engaged with their community and got community members to visit their store. As described elsewhere [21], these data were analyzed and combined with other data sources to produce a highly descriptive narrative known as a case report for each site. These reports served as the unit of analysis for this study. Each report documented the community engagement strategies planned and employed by each store; and while these sections of the report provided data most relevant for this study, the complete case report for each site was analyzed to understand each store’s approach to community engagement within their unique context.

### 2.3. Analysis

We applied Stake’s multiple case study analysis approach [23], moving from within- to cross-case analysis. Analysis began by each site analyzing their multiple data sources to create a within-case report, as described above (additional details are provided elsewhere [21]). These reports were then used in the cross-case analysis, which aimed to maintain the contextual uniqueness of each case while examining for similarities and differences across cases. This phase was led by the lead author (RK). In this phase, each case report was first read in depth to become familiar with the case, and findings relevant to community engagement were extracted. Each case was also rated for the overall utility it would serve in answering the research question, allowing cases with prominent data on community engagement to be identified as well as atypical cases that had uncommon experiences to be flagged. Then, the unique findings to each case went through a series of analytic activities outlined by Stake (e.g., sorting based on similarity, ranking to assess the importance of a finding in answering the research question) [23] that allowed findings to be merged and analysis to move towards generalizing across cases. This set the foundation for developing meaningful assertions and interpretations of the concept of community engagement among food stores in the multi-case study as a whole. Throughout, case reports were re-read, and data checks and confirmations were performed with each site if needed. The process was also supported by a smaller group of co-authors (MW, SJ, JD) that provided weekly feedback on the extracted case findings, merged findings, assertions, and interpretations.

## 3. Results

The seven cases included in this study were located in urban areas across the eastern and midwest regions of the United States (U.S.). Nearly all stores served a largely low-income, Black, Indigenous, and People of Color community. Additional information regarding store characteristics, financial model, and mission has been published [21]. Below, we present our results by first describing the distinct variation in the forms of customer vs. community engagement (Figure 1) observed across sites (Section 3.1), and then highlight five salient strategies from least to most collaborative with latter strategies (Section 3.2) that illustrate the ways stores engaged with their communities.

### 3.1. Customer Engagement vs. Community Engagement

Across the stores in this study, retailers displayed a consistent use of activities that engaged customers, yet fewer stores moved beyond these activities to consistently engage with the surrounding community (Table 1). Receiving customer feedback through multiple channels such as informal conversations during customer check-out, suggestion boxes, and customer surveys were the most common activities of engagement. However, these efforts only allowed retailers to reach customers visiting the store as opposed to a broader population living in the community. Activities that engaged the larger community were most represented across the stores by hosting community events, such as cooking demonstration classes, annual customer appreciation parties, and celebrating a store owner’s birthday.

### 3.2. Strategies of Community Engagement

Five strategies that indicated the different levels of community engagement were identified across sites (Table 2). We present these strategies from least to most collaborative, with latter strategies best representing authentic forms of community engagement. In the following sections, each strategy is described, and nuanced illustrations provided to demonstrate what the strategy looks like in stores, while occasionally drawing comparisons to stores that did not implement the strategy.

#### 3.2.1. Outreach

Outreach describes the channels stores used to establish a flow of communication to inform communities about retailer events and activities [5]. It is recognized as a first strategy towards attaining successful community engagement among participated cases. The source and frequency of communication from retailers to community residents varied. Retailers with frequent communication with the community used different media channels, such as newsletters (Buffalo, Minneapolis, Washington, DC, USA), annual and quarterly reports (Minneapolis, Washington, DC, USA), local radio stations (Buffalo), and social media to highlight sales and recipe videos (Minneapolis, Washington, DC, USA), and were able to effectively reach their target audience. The Boston retailer also reported a successful social media page (with more than 13,000 likes) to communicate information regarding store operations and program initiatives and complemented this with an occasional mailing highlighting store employees and successes to make a case for store donations.

In contrast, some retailers (Chicago, Detroit) were dependent on “word of mouth” marketing and outreach, which owners felt was less effective to reach a wider audience. For Baltimore, a social media page was created but limitedly used (about 1 post per month). The store also planned weekly promotional circulars but did not fully implement due to distribution costs.

#### 3.2.2. Relationship Building with Community through Customer Service and Relations

Fulfilling customers’ needs was a central focus across all participating food retailers in this study. In particular, building a relationship with them, especially through customer service, was recognized as a key step towards also building a relationship with the community. For instance, the Washington, DC, store owner cited how informal conversations with customers can be transformational:


*“Now you’re having a conversation human to human with your customer and it’s no longer this transactional relationship, but it becomes a neighbor-to-neighbor relationship.”*
[Washington, DC]

Further, retailers described different ways they built and maintained these relationships. A store owner at Chicago site emphasized “respect” and “empathy” in customer service, and another retailer at Boston store encouraged his employees to introduce themselves to customers by name. This commitment to connecting with and serving the community was consistently reflected across sites, with many stores’ leadership describing their stores as a “community space” rather than just a ”grocery store”.


*“It’s a church, it’s your local grocery store, it’s your confession center, your child center; this place is more than just fruits and vegetables. Some people just come in and don’t buy stuff; “they just want to come in and talk to us.”*
[Boston]


*“People come to shop and do their grocery shopping, but I think what is unique about our grocery stores is that it’s a community gathering place…usually, it’s like people just hanging out in the aisles talking… We have people who hang out all day in our dining area. I think it’s a place where folks can just be in community.”*
[Minneapolis]

#### 3.2.3. Giving Back to Community

The importance of giving back to the community was reflected across cases as another important strategy to community engagement; yet how stores did this varied widely. The most common way retailers demonstrated this was through hosting free nutrition education classes for the community residents. Other ways reported by retailers included: public health fellowships for the community residents and public health students (Washington, DC, USA) to understand food access in low-income communities, a general staff training for local individuals to develop customer service skills (Washington, DC, USA), and a round-up program to support local organizations through grants and scholarships (Minneapolis).

Giving back to the community also meant having store practices that directly supported the local food system, such as supporting local farmers and/or Black-owned businesses by prioritizing these vendors when procuring products (Buffalo, Minneapolis, Washington, DC). As a stakeholder at the Washington, DC, site mentioned,


*“Our farmers need us to buy food so they don’t go out of business, your community members need food so they’re hungry; like, let’s figure this out. We’re ethically and morally aligned… and it really was wonderful, it really entrenched our relationship for the long term, that we’ll always have something going now.”*
[Washington, DC]

Similarly, a staff member at the Buffalo site reported inviting local business owners and farmers’ markets to participate in a store-hosted community event of celebrating the owner’s birthday annually to get money flowing with people in the community. As he described:


*“The main reason I was thinking about it is because I want to do something to help foster newcomers in our community and then when the pandemic hit, hearing about how my people were struggling, I was like, this would be a great way to get money flowing with people in our community.”*
[Buffalo]

Other retailers also gave back to their communities, but their actions indicated that this was less of a priority. At the Boston, Chicago, and Detroit sites, community events such as annual neighborhood parties to appreciate customers and/or cooking demonstration classes to promote nutrition education were hosted. While these events were specified by some retailers as an attempt to foster engagement with the surrounding community, the infrequency of the events suggested that giving back may have been of less importance.

#### 3.2.4. Partnering with Diverse Community Coalitions

Community coalitions encompass a relationship formed by a group of individuals and/or organizations working collaboratively to achieve a common mission [24] and was consistently recognized as a key characteristic for authentic community engagement across the cases. These coalitions enabled retailers to establish partnerships with organizations working towards a similar goal (i.e., to provide healthy and affordable food to the communities in need). Most retailers engaged with the community through participation in community coalitions; while one site (Baltimore) planned, but never formed, a community coalition (see Table 1).

Retailers that were participating in community coalitions met two criteria: (1) they were actively engaged in diverse partnerships; and (2) the retailer’s motivation to engage in a coalition was altruistic (i.e., to influence the food environment of the community) rather than self-serving (e.g., to increase customer traffic). The first criterion demonstrates the broad support the retailer had from the community, and later helps to explain the retailer’s attitude towards their contribution to the community. For instance, some cases had great diversity in partnering organizations which included faith-based organizations, local media partnerships, public schools, the local library, and academic institutes. In contrast, other sites had community coalitions, but were less diverse, such as having partnerships with mostly health centers (Boston). Retailers that were participating in community coalitions also demonstrated an altruistic motivation to engage in a coalition, through broader missions that focused on communities, rather than just focusing on food availability or improving in-store services to increase customer traffic. For instance, the Minneapolis retailer had the mission statement “To sustain a healthy community that has: equitable economic relationships; positive environmental impacts; and inclusive, socially responsible practices”. To accomplish this mission, along with partnerships with local food growers and community organizations, the retailer was committed to environmental sustainability such that customers received incentives for using reusable bags. One of the community residents confirmed this sentiment by adding,


*“...it’s more than just a grocery store. It’s a statement of our values. It’s a commitment to sustainability in our environment and our food systems. It’s a commitment to support locally grown, locally produced food. It’s a commitment to shared ownership, a different way of being.”*
[Minneapolis]

In some cases, these community partnerships were initiated and facilitated by a store leader’s ties to the community or the community organization reaching out to the store. For instance, in Chicago, the store had the unique position of having a partnership with a well-organized and motivated community organization that had the momentum of initiating efforts to help small food stores in building relationships and improving food access for low-income communities. At the Buffalo site, a Black-owned business, the owner displayed a consistent commitment and a “sense of belongingness” to the community, which was recognized by community coalition stakeholders. As one shared,


*“I feel like our relationship is a little bit different there. He’s constantly trying to expand his reach; he’s constantly trying to support more folks, whether it is through his store or just within the community. I think that’s unique to my relationship with them.”*
[Buffalo]

#### 3.2.5. Promoting Community Inclusiveness and Representation

The final community engagement strategy we identified across the cases was community inclusiveness, including having community representation as part of store decision making. This strategy was also identified as an essential tool towards authentic community engagement. At most sites, such representation was observed among the store management and staff, as employees lived in the same neighborhoods as the stores. This helped each retailer to understand the community culture and further fulfill their needs and preferences, as one owner described,


*“Knowing and having…African American employees, where we are at, are predominantly in the African American community. [These employees] also have provided me input on what would be a product that we should carry.”*
[Chicago]

Some retailers supplemented employee representation with other strategies to include community views and voices beyond their staff, such as conducting listening sessions and/or focus groups to understand the unique and ever-evolving needs of the community served. In Boston, the representation among employees was at the level of store management (but not ownership), which allowed them to have a stronger role in store operations, decisions, and future directions.

At two sites (Minneapolis and Washington, DC, USA), retailers went beyond just understanding the community needs to include community members in a way that allowed them to shape the governance and/or participate in shared decision making at the retailer [5]. In both cases, the retailers had business models (i.e., a cooperative or a social enterprise model) that focused on community inclusiveness in decision making. In the co-op model, used in Minneapolis, the owners were community members rather than investors, and every co-owner (over 22,000) had a single vote for electing the board of directors or other retail changes. To ensure that co-op ownership was accessible, ownership equity is discounted by 80% for low-income and other needs-based residents. The social enterprise model, used in Washington, DC, moved beyond the retail environment and created real social impact through offering additional services and avenues for broader community involvement among the store management team. The retailer has a non-profit community partner as its parent organization which has further partnerships with local universities and youth organizations to engage community residents and youth in the food system through experiential learning opportunities. A majority of the central management and store staff at this site were also local and participated in the store decision-making process of stocking goods.

Across all sites, the effects of having limited community representation in leadership and a commitment to community inclusion was particularly salient at the Baltimore site. While the store hired employees from the neighborhood, the store manager was not a community resident and there was a general lack of commitment to community goals. Further, the store had inadequate community input in store planning and community engagement activities, such as cooking demos. Such little emphasis on fostering community inclusion may have been a contributing factor to the store closing three years after opening. As one employee explained,


*“There were people who said they didn’t, ‘We don’t even know why you were there,’ ‘We didn’t participate in you guys being there,’ ‘we didn’t understand it.’ They pointed to the fact that there are plenty of food sources [in the community] and so they had done their homework too and understood what our premise was for being there, what our mission was, and from their perspective it didn’t fit, so they most certainly didn’t shop there and were critical of our presence from that.”*
[Baltimore]

## 4. Discussion

The present study explored various types of community engagement strategies and activities used by healthy food stores located in lower-income communities. This is the first study, to our knowledge, to specifically examine the ways neighborhood food retailers engage with their communities to address their needs and interests. Across the seven cases, we identified that food retailers implemented a variety of strategies for different purposes, ranging from informing the community about their store to building relationships and collaboration to involving them in decision making. We identified five specific strategies of engagement, including (1) Outreach, (2) Building Relationships through Customer Relations, (3) Giving Back, (4) Partnering with Diverse Community Coalitions, and (5) Promoting Community Representation and Inclusiveness, which increasingly demonstrated retailers’ commitment to collaborate and partner with their community members in authentic and altruistic ways.

The different community engagement activities that food retailers used in this study reached different levels of community involvement and map onto the CDC’s “Continuum of Community Engagement” framework [5]. Most of the neighborhood retailers in this study demonstrated a collaborative degree of community engagement in which, according to the framework, partnerships are formed, communication is bi-directional, and trust is established [5]. In all cases, neighborhood stores were doing more than simple outreach and consultation with communities and attempted to involve them in some way. However, only a few sites were able to involve the community in shared decision making and leadership. The challenges and hesitancy of involving communities in shared decision making has also been observed in other areas of the food environment (e.g., policymaking [25]) and highlights the distinguishing commitment these neighborhood food retailers have made to sharing power and prioritizing the needs and interests of their communities.

We also identified that attaining the highest level of shared leadership and decision making between retailers and communities was best facilitated by the use of a non-traditional business model, such as a co-operative or social enterprise model. The advantages of these models are well established in food retail and other sectors [26,27,28,29,30,31,32]. In the consumer co-op model, as used in the Minneapolis case, co-ownership is embraced among those who consume offered products and/or services [31], and there is growing evidence of its potential in meeting community food needs and improving food access [32]. For instance, Brighter bites, a multi-component school-based intervention to increase consumption of fruits and vegetables among schoolchildren, implemented a food co-op model among parents and found it to be an effective way to empower community members to engage in health promotion [33]. We also found that shared decision making was achieved at retailers through community representation among store employees. However, this was only the case when representation was among store leadership versus staff, underscoring the importance of not just hiring locals but ensuring they have the power and opportunity to shape store decisions.

Partnering with diverse community coalitions was also identified as another important way food retailers fostered effective community engagement. This finding is consistent with literature illustrating that cultivating diverse memberships and encouraging active participation is associated with successful coalitions [34]. Leveraging community coalitions has also been recognized as an essential element for assessing and sustaining the impacts from multilevel retail food interventions [35,36] as well as being an effective approach to addressing pressing community issues such as the obesity epidemic [37]. Retailers that partner with multiple stakeholders have a distinct opportunity to jointly address difficult problems in their communities, including those tied to health as well as a community’s overall social and economic well-being [34].

Our findings help to bring understanding to the commonly used and acceptable community engagement strategies among neighborhood food retailers and thus have several implications for research and practice. While all strategies demonstrated utility in engaging with communities, future research should evaluate which strategies and in which combinations may be most effective to shift communities and retailers to more health promoting community outcomes. In addition, understanding the resources (time, money, social networks) required to develop and sustain strategies at different levels of community involvement and the best ways to access these additional supports could help to inform how to further expand their adoption. Results also provided strong support for retailers to consider non-traditional businesses models, such as cooperatives, that are inherently built upon principles of community inclusiveness and shared decision making. Local policymakers and other city officials may want to consider ways to support and facilitate such business models that prioritize the health, well-being, and needs of their local constituents. Finally, these results may be useful for food retailers to: gauge their current approach to community engagement; evaluate their priorities around serving community interests; and gain insight into the additional strategies required if they wish to build relationships that can effectively address their community’s needs.

### Strengths and Limitations

This study has several strengths and limitations. Strengths of this study included the diversity of participating retailers in terms of location, financial model, store type and size [21], as well as employing a case study approach which allowed for in-depth examination among a varied set of neighborhood retailers. Limitations of this study included the lack of cases from other regions of the U.S., and all were non-chain neighborhood food stores selected for their commitment to making healthy food accessible to low-income communities—both of which may limit the transferability of results to other food retailers. Further, while community member perspectives were integrated into each case via interviews and other sources (e.g., organizational documents, newspaper articles), the study approach primarily centered the retailer’s experiences around community engagement and may have benefited from a fuller incorporation of customer and community perspectives in each case.

## 5. Conclusions

Community engagement has been core to many successful retail food interventions [38,39,40,41] and was recognized as one of the main strategies for retail food business success in this multiple case study project [42]—see John et al. (2022) for further detail. Our study adds to this growing evidence by identifying and describing community engagement strategies that neighborhood food retailers are already utilizing outside the context of a research intervention. We found that the shared decision making is crucial in reaching the highest level of community engagement which further can be accomplished through promoting community inclusiveness in store leadership and non-traditional business models. Other forms of community engagement activities such as outreach, hosting and/or participating in community events, and collaborating with local organizations can also help retailers to progress on the community engagement continuum. Retailers should work to understand their current form of community engagement and how involved the community is in their store decision making if they want their store to attend to community needs. They also need to recognize different priorities which helped shape strategies for improving healthy food access and affordability in their specific context. As these neighborhood retailers demonstrate, when authentic community relationships are prioritized, retailers have the distinct opportunity to learn about the most pressing needs affecting their communities and can begin the work to effectively addressing them together.

## Figures and Tables

**Figure 1 ijerph-19-06986-f001:**
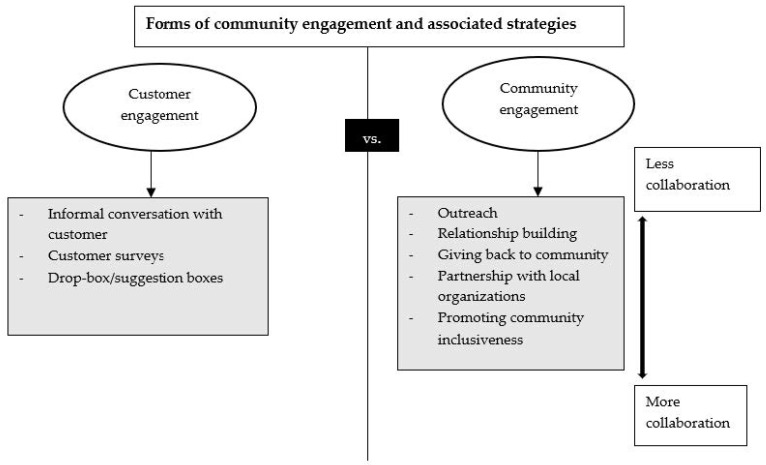
Forms of community engagement in neighborhood food retail stores and associated strategies.

**Table 1 ijerph-19-06986-t001:** Customer and community engagement activities and level of implementation across cases.

	Baltimore, MD	Boston, MA	Buffalo, NY	Chicago, IL	Detroit, MI	Minneapolis, MN	Washington, DC
**Customer engagement**							
Customer feedback (informal conversation/ survey/drop box)	✔	✔✔✔	✔✔✔	✔✔✔	✔✔✔	✔✔✔	✔✔✔
**Community engagement**							
Listening sessions/focus groups	✔	✔✔✔	✔✔✔			✔✔✔	✔✔✔
Media usage	✔✔	✔✔✔	✔✔✔		✔✔✔	✔✔✔	✔✔✔
Community coalitions ^1^	✔	✔✔✔	✔✔✔	✔✔✔		✔✔✔	✔✔✔
Participation in community events	✔	✔✔✔	✔✔✔	✔✔✔		✔✔✔	✔✔✔
Hosting community events	✔	✔✔✔	✔✔✔	✔✔✔	✔✔✔	✔✔✔	✔✔✔
Community representation ^2^	✔✔	✔✔✔	✔✔✔	✔✔	✔✔	✔✔✔	✔✔✔
✔ = Only planned but not implemented ✔✔ = Planned and partially implemented ✔✔✔ = Planned and fully implemented 1. Community coalition refers to the partnerships the store had with other organizations in the community. 2. Community representation refers to the representation among the store staff and/or leadership team (either through employment or ownership).							

**Table 2 ijerph-19-06986-t002:** Community engagement strategies, associated activities, and case sites that demonstrated a use of the strategy.

Strategies	Definition	Example Activities	Case Sites with Evidence of the Strategies
Outreach	Using communication channels to inform communities regarding retailer events and activities	Media usage; annual reports	Boston (MA), Buffalo (NY), Detroit (MI), Minneapolis (MN), Washington (DC)
Relationship building with community through customer service and relations	Developing connections with customers to establish two-way information sharing among retailers and community members	Survey, drop box, focus groups, listening sessions, participation in community events	Boston (MA), Buffalo (NY), Chicago (IL), Detroit (MI), Minneapolis (MN), Washington (DC)
Giving back to community	Supporting community residents and local business to achieve their goals and enhance nutrition education	Hosting community events (e.g., cooking classes); donations to community organizations; prioritizing local vendors; workforce development programs	Boston (MA), Buffalo (NY), Chicago (IL), Detroit (MI), Minneapolis (MN), Washington (DC)
Partnering with diverse community coalitions	Building a partnership with different local community organizations to improve food access in the community	Community coalitions	Boston (MA), Buffalo (NY), Chicago (IL), Minneapolis (MN), Washington (DC)
Promoting community inclusiveness and representation in decision making	Having community members and/or representatives from the community participate in store decision making and future directions	Community representation in management/leadership/ownership; cooperative or social enterprise business model	Boston (MA), Minneapolis (MN), Washington (DC)

## Data Availability

The datasets used and/or analyzed during the current study are available from J.G. (jgittel1@jhu.edu) on reasonable request.

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
