# Peer review of "Forms of Community Engagement in Neighborhood Food Retail: Healthy Community Stores Case Study Project"

_ijerph, 2022, doi:10.3390/ijerph19126986_

Round 1

Reviewer 1 Report

1. Title of Exploring Community Engagement in Neighborhood Food Retail: Healthy Community Stores Case Study Project - suggests a wider range of analyzes and conclusions. It is proposed to consider the approach: forms of engaging.....

2. Abstract should contain information about the data source (researched used).

3. In section 3.1 (lines: 175-176) they indicate the organization of social events. What are these, for example?

4. Conclusions need to be developed.

Author Response

Thank you for your feedback and reviewing our manuscript. Please see the attachment for our responses. 

Reviewer 2 Report

I have one suggestion. If the Authors had photos of the described stores, it would enrich the article. Thank you. The article was very interesting.  This kind of research is very demanding.

Author Response

(The authors gave the same response as above.)

Reviewer 3 Report

The manuscript is very informative and carefully prepared, and the methods and results are very well documented. The subject is original. It describes the importance of the relationship between food retailers with surrounding communities focused on improving community health. It is very similar and essential for obtaining high-quality products when considering the relationship between food processors and raw material producers.

However, the text contains so many details and information that it is difficult to come through.

Beneficial would be the figure, as a model, dynamic model, combining the crucial achievements of the manuscript in areas: of customer engagement vs. community engagement and community engagement strategies.

Based on such a model, it would be more accessible to overwhelming potential limitations as concerns lack of cases from other regions of the USA or cases for other areas resulting from the possible cooperation of the USA and European scientists.

Author Response

(The authors gave the same response as above.)
